# Impacts of Leaf Damage Intensity on Ant–Plant Protection Mutualism and Plant Fitness

**DOI:** 10.3390/plants14060837

**Published:** 2025-03-07

**Authors:** Isabela Cristina de Oliveira Pimenta, Eduardo Soares Calixto, Kleber Del-Claro

**Affiliations:** 1Postgraduation Program in Entomology, Department of Biology, University of São Paulo, Ribeirão Preto 14040-900, SP, Brazil; isalpj@hotmail.com (I.C.d.O.P.); delclaro@ufu.br (K.D.-C.); 2Entomology and Nematology Department, University of Florida, Gainesville, FL 32608, USA; 3Institute of Biology, University Federation of Uberlândia, Uberlândia 38408-100, MG, Brazil

**Keywords:** simulated herbivory, extrafloral nectaries, herbivory, reproductive success, protection mutualism

## Abstract

Herbivores can negatively impact plant reproduction by altering floral traits, pollination, and fruit production. To counteract this, plants developed defense mechanisms, such as the biotic defense resulting from associations with ants. The aim of this study was to investigate whether leaf herbivory at different intensities influences reproductive success and extrafloral nectar secretion patterns in a savanna plant, *Banisteriopsis malifolia* (Malpighiaceae). Plants were subjected to simulated leaf herbivory and divided into three groups: Control (damage < 5%), T15 (15% leaf area removed), and T50 (50% leaf area removed). Assessments continued until fruiting. The findings indicate an increase in extrafloral nectar sugar concentration after simulated herbivory. Increasing foliar damage significantly delayed the time to bloom, decreased the number of inflorescences per plant, and reduced the size of buds and flowers. Foliar damage significantly decreased fruit size. Furthermore, ant foraging was influenced by herbivory, with a predominance of aggressive ants on plants with high levels of damage. Our study shows that varying levels of leaf damage affect extrafloral nectar secretion, ant foraging behavior, and plant reproductive structures. These findings highlight how insect herbivores and the level of damage they cause influence plant fitness and consequently community structure.

## 1. Introduction

Plant–herbivore interactions emerged at the beginning of the Devonian, around 400 million years ago [1]. This antagonistic relationship, which is characterized by insects feeding on plant tissues [2,3], in general has a negative impact on the reproductive success of plants [4]. The extension of herbivore damage is not restricted to leaves, and commonly extends to floral structures [5]. Multiple studies have demonstrated that insect herbivory on leaves can lead to alterations in plant floral traits, potentially reducing photosynthetic area and/or reallocating resources [6,7]. These changes can indirectly influence the behavior of floral visitors, thereby diminishing the plant’s fruit set [8,9,10]. The resulting impacts, whether direct or indirect, can lead to positive, neutral, or negative outcomes for pollinator behavior [8,11,12,13].

Plants have evolved various mechanisms and strategies to minimize or prevent damage from insect attacks. These adaptations result in changes to their primary and secondary metabolism, which can directly affect their reproductive success [11,12,13,14,15,16,17]. These mechanisms can be chemical, physical, phenological, or biological [18,19,20,21]. Chemical defense in plants is exemplified by secondary compounds such as alkaloids, terpenoids, and volatile organic compounds (VOCs) [8,22]. Physical defense is characterized by morphological or mechanical adaptations, such as thorns and trichomes [5,23,24]. Relatedly, plants can escape herbivores by modifying their phenological events, causing a potential mismatch between the peak of herbivore abundance and the production of specific tissues [20,25,26]. Additionally, plants can associate with natural enemies of their herbivores, such as ants, wasps, and spiders, establishing a protective mechanism called biotic defense [27,28,29]. Many plants, especially angiosperms, present structures to provide shelter and/or food resources as is the case of extrafloral nectar produced by extrafloral nectaries (EFNs) [30,31]. These defenses can work alone or in combination to reduce the damage caused by insects. Even so, some insect herbivores can surpass these defenses and inflict different levels of damage to plants, which may induce certain responses [32,33].

One type of induced response by EFN-bearing plants is the induction of extrafloral nectar production, which can alter ant foraging patterns [31,34]. For example, studying the induced response of EFNs of *Qualea parviflora* (Vochysiaceae), Raupp et al. [34] found that the type of leaf damage (chewing or puncturing) can influence the outcomes of ant–plant mutualism. Furthermore, the levels of induction of EFNs and the response of ants to this damage are directly relative to the level and location of damage on *Qualea multiflora* (Vochysiaceae) [35]. However, herbivory does not only affect the induced defenses of plants; it can trigger different responses, either directly (e.g., alteration of the floral characteristics of the plant) or indirectly (e.g., alteration in pollination patterns), which result not only in losses to the plant [36,37], but also in benefits [8,38]. Few studies have explored how the intensity of leaf damage can alter EFN production, ant behavior, and plant reproductive success. Studying the effects of leaf damage intensity on EFN production, ant behavior, and plant reproductive components provides important insights in how varying levels of herbivory influence plant defenses and mutualistic interactions. This improves our understanding of the balance between costs and benefits of induced responses, ultimately affecting ecosystem dynamics.

Here, we evaluated whether leaf herbivory at different intensities on a Brazilian savannah tree bearing EFNs, *Banisteriopsis malifolia* (Malpighiaceae), influences extrafloral nectar secretion patterns, foraging and activity of associated ants, and the reproductive success of the plant. We conducted a field experiment using different levels of artificial damage to plants to assess how this plant responds EFN quality, ant foraging patterns, and plant reproductive success. The main predictions are: (i) low levels of leaf damage (<5% of total leaf area) are unlikely to generate significant variations in the factors studied (EFN volume and sugar concentration, ant behavior, reproductive components); (ii) moderate levels of damage (15% of total leaf area) are expected to elicit a stronger induced response compared to minimal damage, leading to positive changes in ant foraging behavior (increasing ant attraction to the plants) and enhancing reproductive components of the plant. This should also result in greater EFN volume and sugar concentration [31] and (iii) high levels of damage (50% of total leaf area) will have a dual effect: positively influencing ant foraging behavior while negatively impacting the plant itself by substantially reducing its photosynthetic area. This reduction in photosynthesis can limit the allocation of resources for reproduction, but is still expected to produce higher EFN volume and sugar concentration compared to lower damage levels [6].

## 2. Results

### 2.1. Nectar Volume, Sugar Concentration, and Ant Abundance and Richness Under Different Damage Treatments

Increased leaf damage did not influence the volume of extrafloral nectar produced between treatments (χ^2^ = 0.49, *p* = 0.7819, R^2^m = 0.008, R^2^c = 0.23), but it had a positive effect on the sugar concentration of the nectar (χ^2^ = 6.99, *p* = 0.03, R^2^m = 0.13, R^2^c = 0.13, Figure 1). T15 plants had a sugar concentration 1.45 times higher (mean = 36.0 [95% confidence interval (CI): 28.2, 43.7]) than plants without simulated herbivory (mean = 24.8 [95% CI: 19.0, 30.5]); T50 plants (mean = 34.2 [95% CI: 26.5, 42.0]) did not differ significantly from either treatment (Figure 1).

A total of nine species of ants were observed foraging on the plants when the nectar was collected. Of these, six species were seen in the Control and T15 treatments and only two species were found in T50 (Table 1). Plants without simulated herbivory had many ants of the species *Ectatomma tuberculatum* (59%), as did T50 plants (65%). Unlike the T15 plants, which did not have ants of this species at the time of the evaluation, we did find ants of other species, such as *Wasmannia* sp. (44%), which had the highest relative abundance. In addition, we observed that only two species of ants appeared on T50 plants at the time of the assessment: *E. tuberculatum* and *Camponotus* sp.

Since treatment T15 influenced the sugar concentration of the nectar, we analyzed the influence of the interaction between the treatments and the sugar concentration on the abundance and richness of ants. There was no influence of the interaction between treatments and the sugar concentration of the extrafloral nectar on ant abundance (χ^2^ = 1.335, *p* = 0.512). However, when analyzed separately, both treatments (χ^2^ = 11.650, *p* = 0.002) and sugar concentration (χ^2^ = 4.053, *p* = 0.044; Figure 2) had significant effects on ant abundance, which suggests that ant foraging is influenced by the herbivory presented in the treatments. It was found that for every one unit increase in nectar sugar concentration, there is an increase of 1.03 ants on the plant (mean = 1.24 [95% CI: 0.733, 2.09]). Similarly, treatment (χ^2^ = 3.618, *p* = 0.1637), sugar concentration (χ^2^ = 0.3167, *p* = 0.5736), or their interaction (χ^2^ = 0.7989, *p* = 0.6707) did not have a statistically significant effect on ant richness on plants.

As the previous analysis showed that the treatments influence the abundance of ants on the plant, an interaction was conducted between the treatments and the abundance of ants in the total number of samaras produced. There was no influence of the interaction between treatments and the abundance (χ^2^ = 0.4452, *p* = 0.5046) of ants on the plant on the total number of samaras produced, nor was there any significant influence of ant abundance (χ^2^ = 0.3955, *p* = 0.5294). However, there was an influence of treatments on the total number of samaras produced (χ^2^ = 4.0751, *p* = 0.04; Figure 3a). Neither richness (χ^2^ = 0.004, *p* = 0.9468) nor the interaction between treatments and ant richness (χ^2^ = 0.1397, *p* = 0.7085) influenced the total number of samaras produced, only the treatments (χ^2^ = 3.8163, *p* = 0.05; Figure 3b).

### 2.2. Impact of Simulated Herbivory on Flowering Time and Number of Inflorescences Produced

Simulated herbivory significantly delayed the appearance of inflorescences (χ^2^ = 63.21, *p* < 0.001, R^2^m = 0.52, R^2^c = 0.52, Figure 4a) and the first flowers (χ^2^ = 42.04, *p* < 0.001, R^2^m = 0.43, R^2^c = 0.43, Figure 4b). T15 (37.1 [95% CI: 33.6, 40.7]) and T50 (45.3 [95% CI: 41.5, 49.2]) plants took on average 13 and 21 days longer, respectively, than control plants (24.4 [95% CI: 20.7, 28.1]) to produce inflorescences (Figure 4a). In addition, T50 plants took around eight days longer than T15 plants to start producing inflorescences, indicating that as damage increased, so did the time taken to produce inflorescences. T15 plants (mean = 84.4 [95% CI: 79.5, 89.3]) and T50 plants (mean = 92.0 [95% CI: 86.5, 97.5]) also took longer to produce their first flowers than Control plants (mean = 69.0 [95% CI: 64.0, 73.9], Figure 4b), around 15 and 23 days longer, respectively. T50 plants took around eight days longer than T15 plants to produce their first flowers. The results showed a significant difference between the treatments in the total number of inflorescences produced per plant (χ^2^ = 9.4277, *p* = 0.008, R^2^m = 0.13, R^2^c = 0.13, Figure 4c). Control plants (mean = 18.33 [95% CI: 13.29, 23.4]) produced 1.85 times more inflorescences than T15 (mean = 9.90 [95% CI: 4.86, 14.9]) and 2.39 times more than T50 (mean = 7.65 [95% CI: 2.04, 13.3]).

Plants with simulated herbivory (T15, T50) developed smaller buds and flowers than plants without simulated herbivory (Control) (Table 2, Figure 5). Bud height was 1.04 and 1.09 times lower in T15 and T50, respectively, than in Control (Figure 5a). The diameter of the buds was 1.07 and 1.05 times smaller in plants with simulated herbivory (T15 and T50, respectively) than in plants without simulated herbivory (Control) (Figure 5b). The width of the flowers was 1.13 and 1.10 times smaller in the treatments (T15 and T50, respectively) than in the Control (Figure 5c). In addition, we observed that bud height was lower in T50 (effect.size: Control—T50 = 1.448; T15-T50 = 0.834, Table 2) and diameter was lower in T15 (effect.size: Control—T15 = 0.425; T15-T50 = −0.130, Table 2) when comparing the treatments. Flower width was smaller in T15 (effect.size: Control—T15 = 1.165; T15-T50 = −0.221, Table 2), while flower length did not differ between treatments (Figure 5d).

Simulated herbivory significantly influenced the size of the inflorescences produced. T50 plants produced inflorescences 6.23 times smaller than Control plants and 8.54 times smaller than T15 plants; Control and T15 plants did not differ (Table 2, Figure 6a). The number of flowers produced per inflorescence was also lower in T50. T50 plants produced 7.63 times fewer flowers per inflorescence than Control plants; when analyzing plants with 15% simulated herbivory, we observed that the number of flowers produced did not differ from the other treatments, as there was an overlap of values (Table 2, Figure 6b). When we consider the size of the inflorescence in flower production, we see that T15 plants differ from T50; however, Control plants do not differ from either of the two treatments with simulated herbivory (Table 2, Figure 6c).

### 2.3. Impact of Leaf Damage on Reproductive Success

Simulated herbivory did not influence the natural weight (χ^2^ = 1.8647, *p* = 0.3936) and dry weight (χ^2^ = 2.1033, *p* = 0.3494) of the samarids, but it did significantly influence the size of the samarids (χ^2^ = 16.262, *p* < 0.001, R^2^m = 0.20, R^2^c = 0.55, Figure 7a). Plants with leaf herbivory (T15: mean = 1.95 [95% CI: 1.67, 2.24]; T50: mean = 1.83 [95% CI: 1.52, 2.15]) produced 1.33 and 1.42 times smaller fruit (T15 and T50, respectively) compared to Control plants (mean = 2.60 [95% CI: 2.33, 2.88]). When analyzing the samarids produced per flower (χ^2^ = 7.3931, *p* = 0.02, R^2^m = 0.08, R^2^c = 0.40), we observed that T15 plants (mean = 1.330 [95% CI: 1.02, 1.64]) produced more samarids than T50 plants (mean = 1.094 [95% CI: 0.75, 1.43]) and Control (mean = 0.775 [95% CI: 0.50, 1.04]; Figure 7b). The number of samaras produced per inflorescence (χ^2^ = 4.0524, *p* = 0.13) did not differ between treatments.

## 3. Discussion

Our main hypothesis of the study was corroborated, pointing to different levels of leaf damage influencing EF nectar quality, ant foraging patterns, and plant reproductive success. These outcomes reinforce the argument that EFNs can act as induced defense and shows how plants can differently allocate resources to indirect induced defenses according to the structure (vegetative or reproductive tissue) and severity of herbivory damage [5] (graphical abstract).

The increase in 15% of leaf herbivory induced different responses in the plants, with a greater investment in increasing the sugar concentration of the EFNs [35]. Although there was no statistical difference in sugar concentration between the Control and T50 plants, T50 plants produced 1.37 times more sugar than the Control plants, which may explain why only more aggressive ants appeared in T50 compared to the other treatments (Table 1) [37]. Ant protection can reduce herbivory, allowing the plant to allocate more resources to its growth and reproduction, resulting in increased production of leaves, flowers, fruits, and seeds. There was no increase in nectar volume in any of the treatments. Extrafloral nectar production is a complex physiological mechanism that can vary among plants, plant parts, and even between young and old leaves [34,35,39]. In general, *B. malifolia* appears to produce a low volume of nectar, as shown by Calixto et al. [35]. T15 plants had a higher sugar concentration than the Control, while T50 was similar to the other treatments. This strong reduction in photosynthetic area can limit the production and allocation of resources to the production of sugar. Alternatively, it may indicate that the plant is reallocating resources toward leaf tissue growth to enhance resource availability for reproduction [6,35]. This result contrasts with other studies where the sugar concentration of extrafloral nectar increases with the intensity of damage to the plant [34,35], suggesting that the induced response to herbivory among EFN-bearing savanna plants may be species-specific [40,41].

Different intensities of leaf damage affect the plant’s floral traits, delaying the production time of inflorescences and flowers, producing fewer inflorescences per plant and smaller buds and flowers, as in Mothershead and Marquis [42], who manipulated leaf herbivory in *Oenothera macrocarpa* (Onagraceae) and measured the effects of the damage on its floral characteristics. They found that plants with a greater loss of leaf area produced fewer flowers than plants with natural leaf damage. This is due to a decrease in photosynthates by the plant and potential reallocation of resources, resulting in less investment of resources for reproduction and development in T15 and especially T50 plants [43]. Plants with high levels of leaf damage (T50) produced smaller inflorescences, buds, flowers, and fruit, in addition to producing few inflorescences (max. 3 per plant). Plants with intermediate levels of damage (T15) also had a delay in the production of their inflorescences; however, they produced more flowers per inflorescence and larger inflorescences than T50. These results suggest that there is compensation on the part of the plant, due to the level of damage more accentuated than 5% [11,44]. A greater decrease in the plant’s photosynthetic area impaired its growth and reproduction [6]; T50 plants produced smaller inflorescences and fewer flowers per inflorescence. These results suggest that the plant has possibly reallocated resources; by reallocating its resources, the plant may lack energy to produce flowers and inflorescences, generating changes in the size and shape of the plant’s floral traits [45,46].

The plant’s reproductive success was also affected by the different levels of leaf damage, producing fewer and smaller fruits per flower. T50 plants produced fewer fruits per flower than plants from the other treatments, which can be explained by the low amount of resources produced due to the low photosynthetic area [42]. The greatly reduced photosynthetic area likely also constrained the sugar concentration of extrafloral nectar in plants with 50% damage. In addition, there may have been an impact of the low production of floral resources on pollinator attractiveness and a potential impact of ants on pollinators [37,47,48,49].

Our results highlight the importance of understanding how different levels of leaf damage influence EF nectar production, ant foraging behavior, and plant reproductive success. By showing that intermediate herbivory levels (T15 plants) increase EF nectar sugar concentration, our results reinforce the idea that EFNs act as an induced defense, with plants differentially allocating resources based on the severity of herbivory. The findings suggest that plant responses are highly context-dependent, varying with both the intensity and location of damage (i.e., whether it occurs in vegetative and/or reproductive parts), which can ultimately affect the balance between defense and reproduction [35]. The T15 plants are characterized by a significant allocation of EFNs (extrafloral nectaries) as part of an indirect defense mechanism, aimed at mitigating herbivore pressure. In contrast, the T50 plants allocate the majority of resources to reproduction, dedicating the remaining energy to defense, which results in negative impacts due to the high level of damage incurred. The difference in allocation strategies between these two plant categories highlights that the cost of allocating resources to indirect defenses is beneficial only under intermediate levels of damage. This dynamic has important implications for plant tolerance, as it reflects the extent to which a plant can endure damage without compromising essential functions, such as reproduction and defense. These results provide insights into the species-specific nature of induced defenses in EFN-bearing plants and emphasize the importance of studying plant–herbivore interactions to better understand how plants manage limited resources under stress.

## 4. Materials and Methods

### 4.1. Study Area and Species Studied

The field study was conducted in the Reserva Ecológica do Clube de Caça e Pesca Itororó de Uberlândia (CCPIU—48°17′27.0″ W; 18°58′30.6″ S), in Uberlândia-MG, Brazil, from January to June 2021, where the main physiognomy is cerrado sensu stricto, composed of trees up to 10 m apart and interspersed with shrubs and grass. The area is also characterized by the presence of vereda, wet field, mesophytic forest, and gallery forest [50]. The climate in the region is marked by two well-defined seasons, a dry and cold one from April to September and a humid and hot one from October to March [51].

*Banisteriopsis malifolia* (Nees and Mart.) B. Gates is a shrub of the Malpighiaceae family, common in shrubby formations of the Cerrado. It has sericeous branches and opposite leaves. The leaves have a pair of EFNs at the base near the petiole, one on each side of the central vein. Each bract had a pair of active EFNs at its base in the abaxial part, one on each side of the central vein. Its inflorescences can be terminal or axillary, with persistent bracts, generally oblong, and sessile pedicels. The flowers contain sepals, their petals are white or pink, turning pale yellow when senescent. The fruits are of the samarid type with greenish color, sericeous and glabrescent towards the apex. Its flowering and fruiting occur from February to October [29,52]. This species is very abundant in the study area and has interactions with ants mediated by EFNs [29].

### 4.2. Experimental Design

At the end of January 2021, 75 plants with similar phenological characteristics (e.g., size and the presence of only branches with young and intermediate leaves), were selected in a large area, at least five meters apart from each other. These plants were randomly divided into three groups of 25 plants each: Control—no experimental manipulation was done and therefore leaf damage was natural (<5% of total leaf area); T15—15% of the apical area of all leaves of all plants was removed by simulated herbivory by cutting; T50—50% of the area of all leaves of all plants was removed also by cutting. The percentage of leaf damage was done by visual percentage estimation, by quadrants. We divided the leaf into four, where each part was equivalent to 25%, in this way we were able to estimate the percentage of herbivory of the leaves. Under natural conditions, the leaf herbivory rate of this species averages 3.48% (standard deviation: 2.01; n = 10 plants). Thus, to evaluate the effects of different levels of herbivory on plants and associated ants, we increased the damage values by approximately 5-fold (group T15) and 15-fold (group T50). Values of 15% herbivory can be seen regularly in the field, while values of 50% are seen rarely (ESC personal communication). The cuts were made on all leaves present on the plant, regardless of ontogenetic stage. However, most leaves were at intermediate or fully expanded, mature stages. The cuts were made with a pair of blunt scissors, always transversely in relation to the leaf length, i.e., in the middle of the leaf lamina at T50 and at the tip of the leaf lamina at T15 (Figure 8). New cuts were made as the plant presented leaf budding, which occurs until the beginning of the reproductive stage, in mid-March [29]. The cuts were always made in the morning between 8:00 a.m. and 12:00 p.m.

### 4.3. Extrafloral Nectaries and Ants

The productivity of EFNs was evaluated to verify their response to herbivory levels, and to correlate the abundance and richness of ants with the quantity (volume) and/or quality (sugar concentration) of nectar. To this end, during the floral bud production phase (mid-April), three bracts were selected, one on each inflorescence. The two EFNs were washed with distilled water, dried with filter paper, and bagged with a voile bag [31]. This procedure was done in the morning, between 8:00 and 11:00 a.m., and the nectar was not evaluated until 24 h later. The volume of the nectar from both EFNs were measured using a 5 μL graduated microcapillary, and the percentage in Brix of sugar was measured with a manual refractometer (model Eclipse) with measurement up to 50%. During the extrafloral nectar assessment, ant abundance and richness were quantified across the entire plant. To ensure accuracy and avoid counting the same ants multiple times, the plants were divided into three sections: upper, middle, and lower thirds. This approach allowed for a systematic and non-redundant ant count. The ants found were collected, transported to the laboratory, and identified using dichotomous keys.

### 4.4. Floral Traits

The variation in floral traits of each plant was evaluated. For this, five buds were selected from one inflorescence per plant, which were bagged with a voile bag. Then, at the pre-anthesis stage, the height and diameter of each bud was evaluated (Figure 9A). The buds were bagged again to prevent herbivore attack, and analyses were resumed after anthesis. After anthesis of the five selected buds, the size (width and length) of the flowers was analyzed (Figure 9B). Both were measured with a digital caliper. The total number of inflorescences per plant and the number of flowers per inflorescence in three previously selected inflorescences were quantified. The total number of inflorescences per plant was counted at the peak of flowering of the plant, which occurs in May, as well as the number of flowers per inflorescence. In previous field observations, it was noted that the flowers do not last longer than three days, and at the end of the first day they already present changes in color and morphology of the structures. Therefore, all buds and flowers were quantified without replication.

### 4.5. Reproductive Success

To evaluate the reproductive success of the plants, the number of samaras produced per flower in the three inflorescences previously selected for the study of floral traits was evaluated, and at least five samarids per inflorescence were collected for size and weight evaluation. Each flower may produce 0 to 3 samarids, and each samarid contained a seed (Figure 9C). The size of the samarids was determined through photos, which were evaluated in the ImageJ program (1.53k; Java 1.8.0), and the weight was measured on a digital analytical scale with four digits (Uni Bloc/ATY224), before and after being dried in an oven (24 h period at 55 °C).

### 4.6. Statistical Analysis

All analyses were performed using R 4.1.1 software [53] at 5% probability. The following packages were used in the analyses: ‘glmmTMB’ [54], ‘car’ [55], ‘emmeans’ [56], ‘lme4’ [57], and ‘multcomp’ [58].

#### 4.6.1. Nectar Volume and Sugar Concentration, and Ant Abundance and Richness

To assess whether the volume and sugar concentration of nectar are influenced by the intensity of leaf damage, a generalized linear mixed model (GLMM) with a negative binomial distribution was conducted, where the volume or sugar concentration of nectar was considered the response variable, and the treatments the fixed effect. The individual plants were adjusted as a random effect. To evaluate the model’s explanatory power, we calculated both marginal and conditional R^2^. The marginal R^2^ represents the proportion of variance explained only by the fixed effects, while the conditional R^2^ indicates the variance explained by both fixed and random effects. An analysis was conducted using only the sugar concentration of the nectar to analyze the effect of the interaction between treatments and sugar concentration on the abundance and richness of ants. To do this, a generalized linear model (GLM) with a negative binomial distribution was fitted in order to control data overdispersion. The abundance or richness of ants was set as the response variable, and the treatments, sugar concentration, and interaction between treatments and nectar sugar concentration were set as the fixed effect. In addition, in order to carry out this analysis, we had to average the sugar concentration of the nectar per plant, due to the fact that we had more than one value of data collected for EFNs and only one value of data collected for ants. Finally, to analyze the influence of the interaction between the treatments and the abundance and richness of ants on the total number of samaras produced, a GLM with a negative binomial distribution was fitted. The total number of samaras produced was set as the response variable, and abundance, richness, and the interaction between treatments and abundance and richness were set as fixed effects. The plants were adjusted as a random effect.

#### 4.6.2. Flowering Time and Number of Inflorescences Produced

To see if simulated herbivory interfered with the time taken for inflorescences and flowers to appear, a GLM with a Gaussian distribution was carried out, where the days to inflorescence appearance and anthesis of the first flower after inflorescence appearance were considered the response variable and the treatments the fixed effect. To find out if there was a difference between the treatments in the total number of inflorescences produced per plant, a GLM with a Gaussian distribution was used, where the total number of inflorescences was considered the response variable and the treatments the fixed effect. To check whether simulated herbivory influenced each of the plant’s floral traits (bud height and diameter, flower length and width, inflorescence size, number of flowers produced per inflorescence, and flower production as a function of inflorescence size), a GLMM was conducted with error distributions according to the type of response variable. Each floral trait was adjusted as a response variable, the treatments were adjusted as fixed effects, and the individual plants as a random effect controlling the spatial dependence of the data.

#### 4.6.3. Reproductive Success and Plant Fitness

To evaluate the components related to the plant’s reproductive success (natural weight, dry weight, and area of samarids, fruit formed by flowers and fruit formed by inflorescence), a GLMM was conducted with error distributions according to the type of response variable. Each component was adjusted as a response variable, the treatments were adjusted as fixed effects, and the individual plants as a random effect controlling the spatial dependence of the data.

## 5. Conclusions

The results of this study demonstrate that the intensity of foliar damage directly affects extrafloral nectar sugar concentration, the foraging behavior of associated ants, and the reproductive success of *Banisteriopsis malifolia*. Intermediate levels of damage (T15) induce a higher sugar concentration in extrafloral nectar, while severe damage (T50) negatively impacts reproductive performance. This suggests that plants allocate resources strategically, adjusting their responses according to stress intensity. Thus, the relationship between defense and reproduction is dynamic and context-dependent, influencing the plant’s tolerance to foliar herbivory. These findings highlight the importance of studying plant–herbivore interactions to understand how plants manage limited resources under environmental pressure.

## Figures and Tables

**Figure 1 plants-14-00837-f001:**
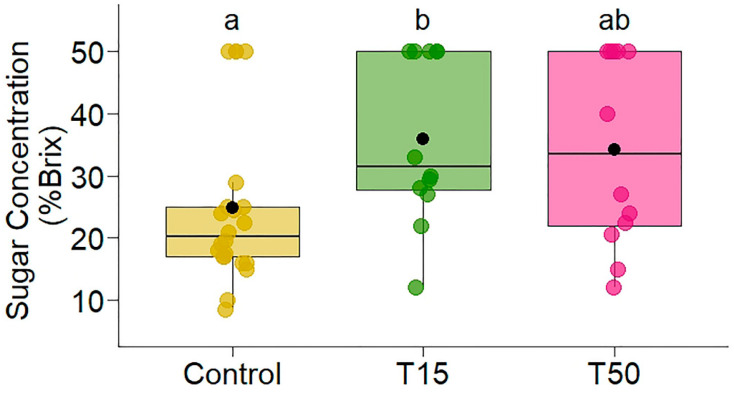
Variation in the sugar concentration of the extrafloral nectar produced by the extrafloral nectaries in *Banisteriopsis malifolia* (Malpighiaceae) plants under two conditions: without simulated herbivory (Control) and with simulated herbivory (T15, T50). T15—plants with 15% simulated herbivory; T50—plants with 50% simulated herbivory. Different letters represent statistical difference by the estimated marginal mean. The figures display boxplots with raw data represented as colored points and the mean indicated by a black point.

**Figure 2 plants-14-00837-f002:**
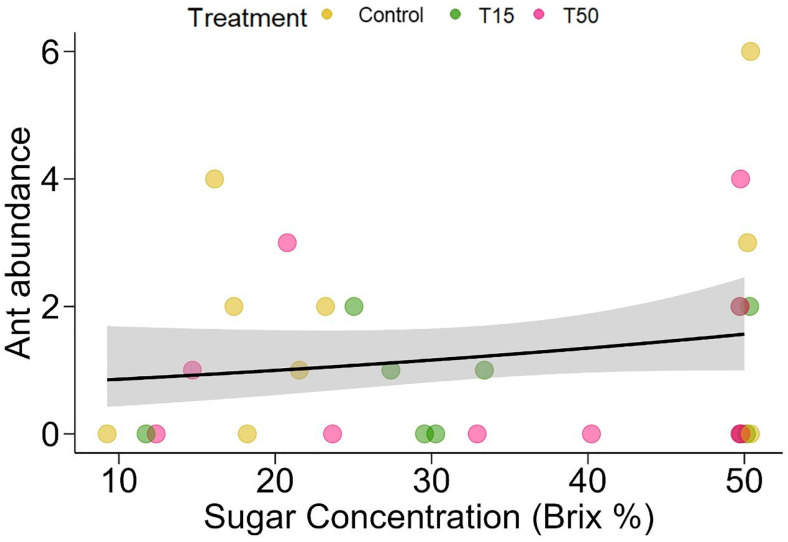
Interaction between treatments and sugar concentration of extrafloral nectar produced by extrafloral nectaries on ant abundance in *Banisteriopsis malifolia* (Malpighiaceae) plants without (Control) and with simulated herbivory (T15, T50). The data points represent raw data.

**Figure 3 plants-14-00837-f003:**
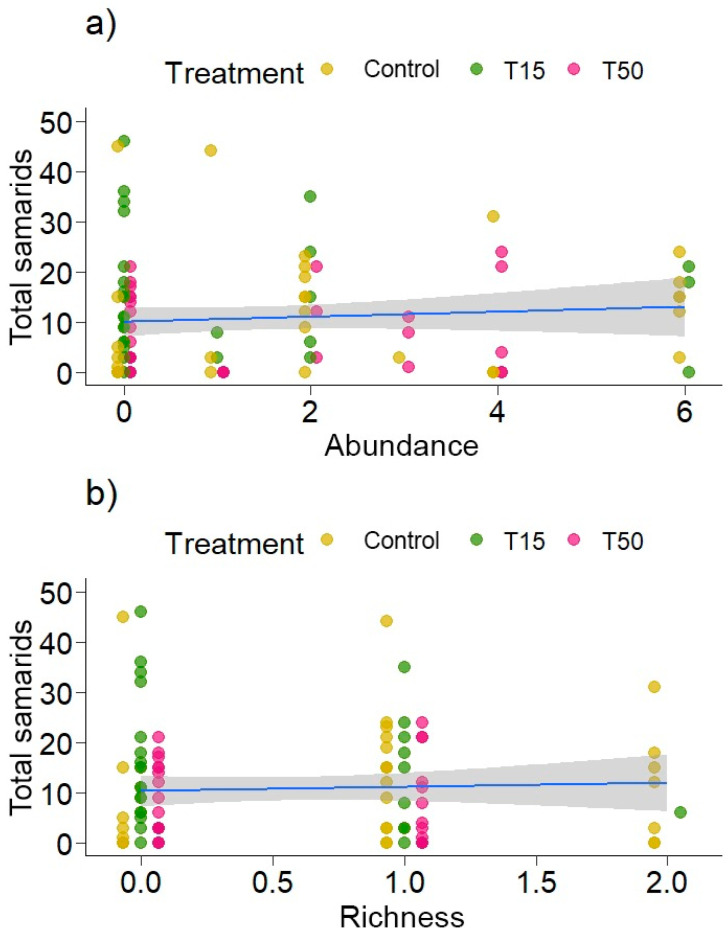
Interaction between treatments and (**a**) abundance and (**b**) ant richness in relation to the total number of samaras produced, in *Banisteriopsis malifolia* (Malpighiaceae) plants, under control conditions (no simulated herbivory) and with simulated herbivory (T15, T50). Data points represent raw values.

**Figure 4 plants-14-00837-f004:**
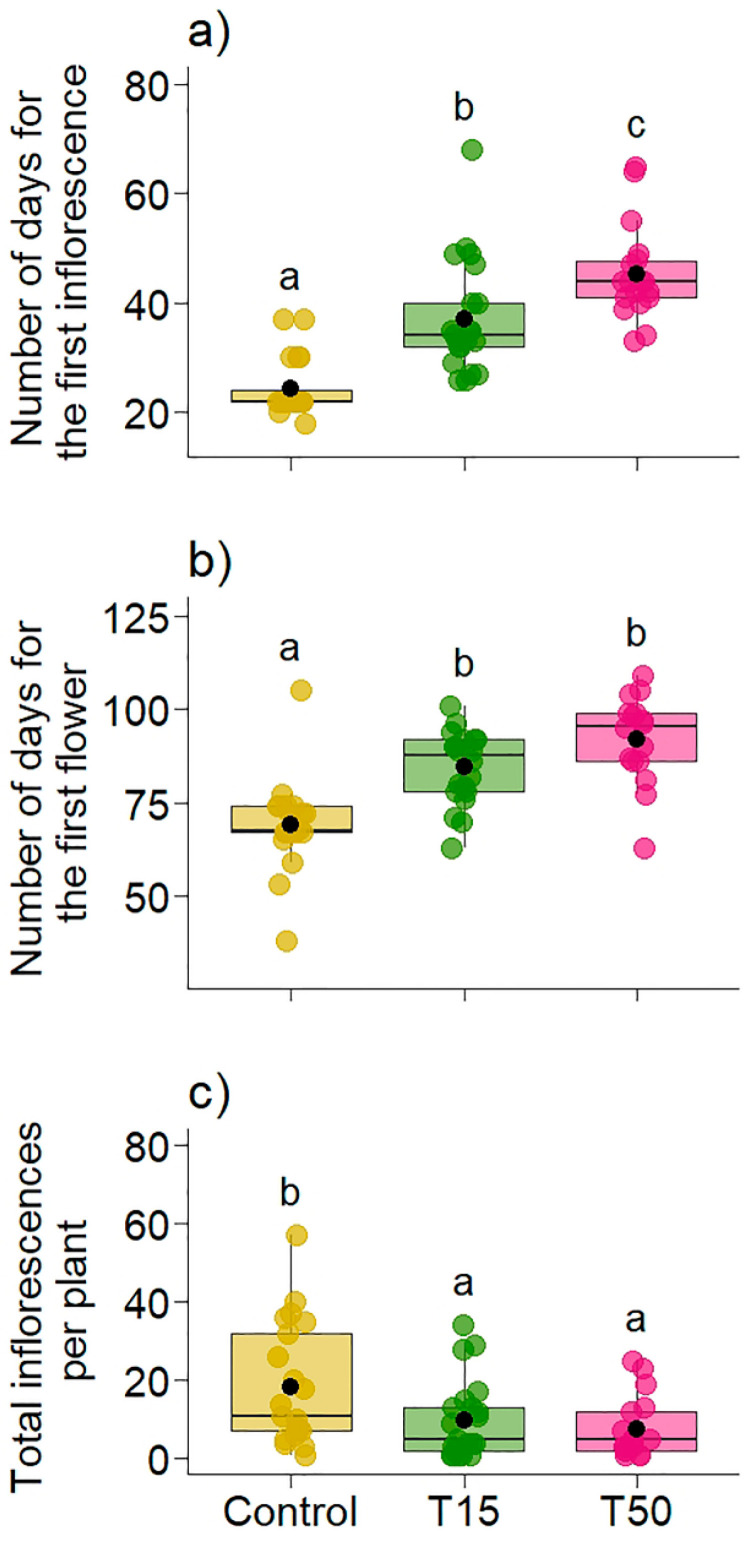
Variation in the (**a**) difference (in days) between simulated herbivory and the appearance of the first inflorescence, (**b**) difference (in days) between simulated herbivory and the blooming of the first flower, and (**c**) total number of inflorescences per plant in *Banisteriopsis malifolia* (Malpighiaceae) plants under two conditions: without simulated herbivory (Control) and with simulated herbivory treatments (T15 and T50), with 25 replicates per treatment. Different letters in (**a**–**c**) represent statistical difference by the estimated marginal mean. The figures display boxplots with raw data represented as colored points and the mean indicated by a black point.

**Figure 5 plants-14-00837-f005:**
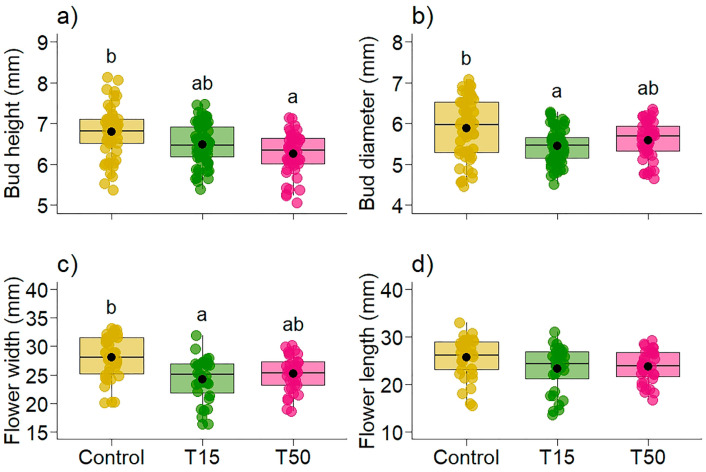
Variation in (**a**) bud height, (**b**) bud diameter, (**c**) flower width, and (**d**) flower length (in millimeters) in *Banisteriopsis malifolia* (Malpighiaceae) plants under two conditions: without simulated herbivory (Control) and with simulated herbivory treatments (T15 and T50), with 25 replicates per treatment. Different letters in (**a**–**c**) represent statistical difference by the estimated marginal mean. The figures display boxplots with raw data represented as colored points and the mean indicated by a black point.

**Figure 6 plants-14-00837-f006:**
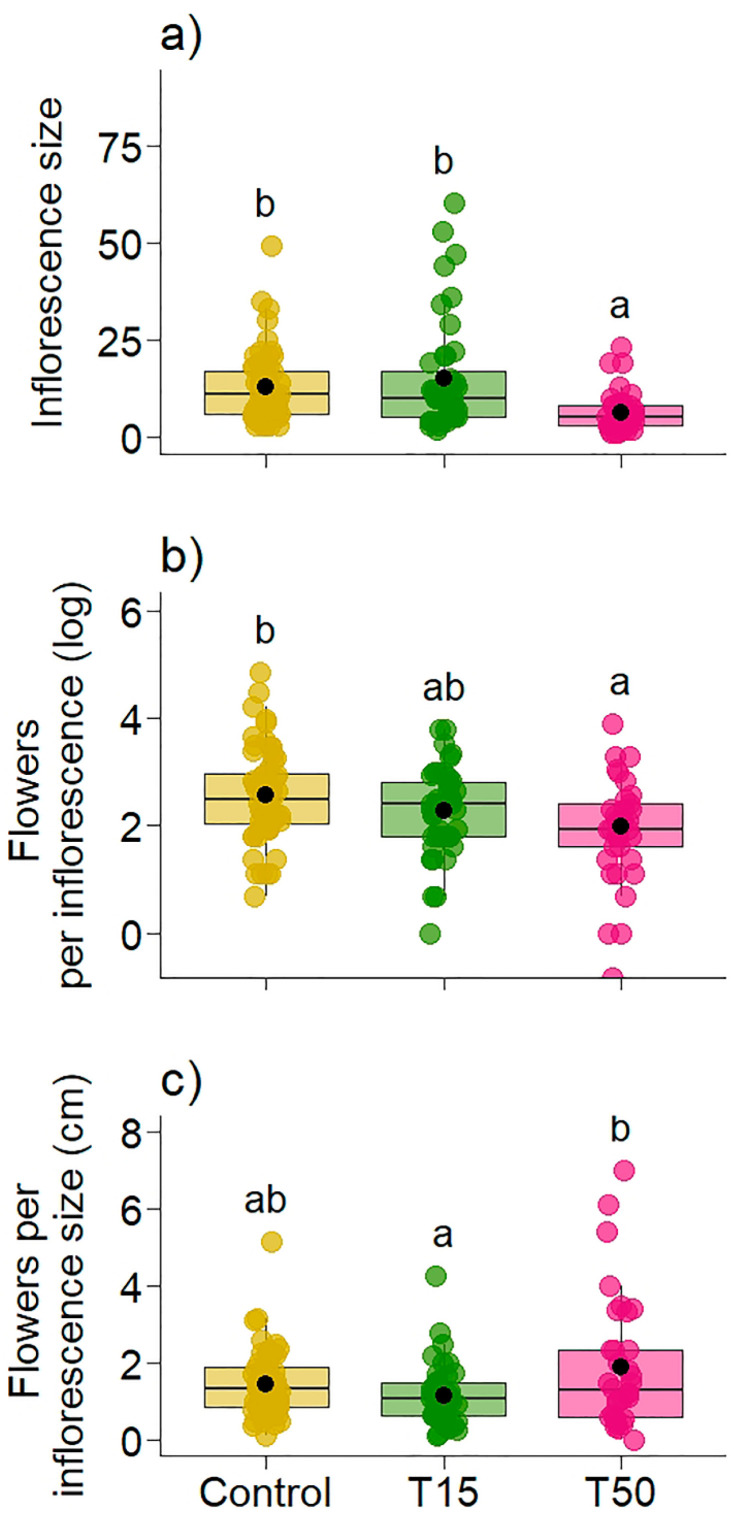
Variation in (**a**) inflorescence size (in centimeters), (**b**) log of the number of flowers per inflorescence, and (**c**) flowers produced as a function of inflorescence size in *Banisteriopsis malifolia* (Malpighiaceae) plants under two conditions: without simulated herbivory (Control) and with simulated herbivory treatments (T15 and T50), with 25 replicates per treatment. Different letters in (**a**–**c**) represent statistical difference by the estimated marginal mean. The figures display boxplots with raw data represented as colored points and the mean indicated by a black point.

**Figure 7 plants-14-00837-f007:**
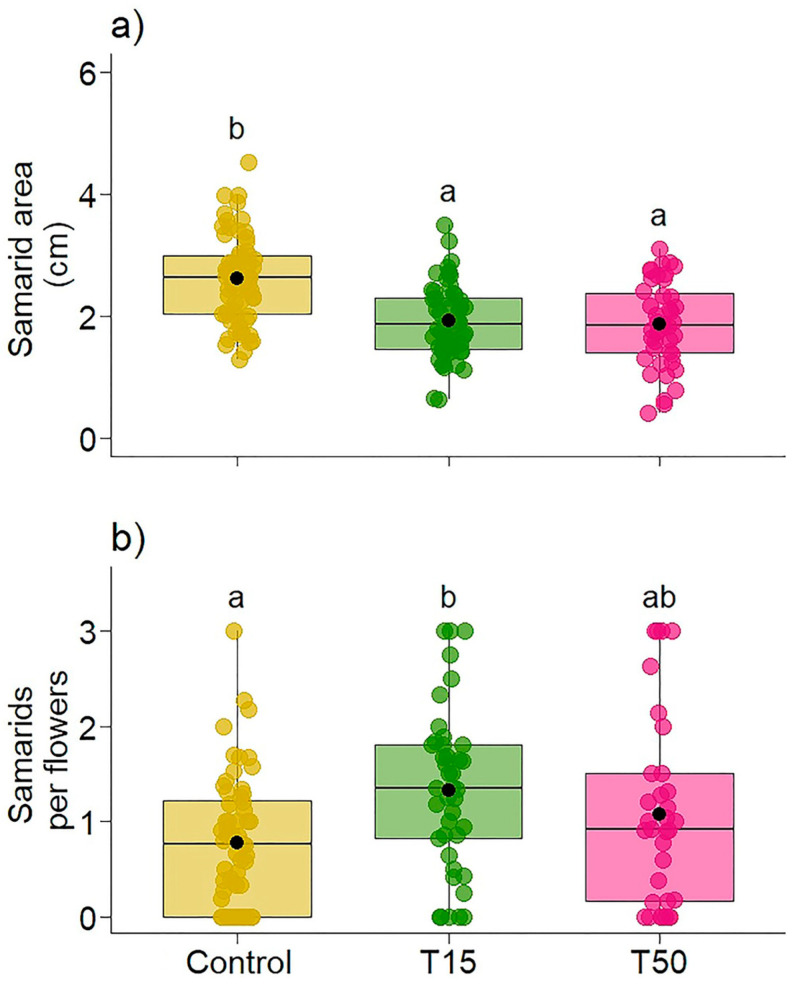
Variation in (**a**) the area of samarids collected (measured in centimeters) and (**b**) the number of samaras produced per flower in *Banisteriopsis malifolia* (Malpighiaceae) plants under two conditions: without simulated herbivory (Control) and with simulated herbivory treatments (T15 and T50), with 25 replicates per treatment. Different letters represent statistical difference by the estimated marginal mean. The figures display boxplots with raw data represented as colored points and the mean indicated by a black point.

**Figure 8 plants-14-00837-f008:**
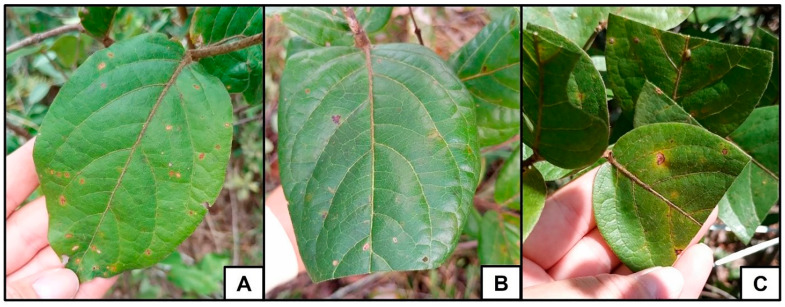
Treatments with simulated leaf herbivory in *Banisteriopsis malifolia*. (**A**) Control Group—leaves without any experimental manipulation, leaf damage less than 5%. (**B**) Group T15—leaves with 15% damage. (**C**) Group T50—leaves with 50% damage. Photos by the author.

**Figure 9 plants-14-00837-f009:**
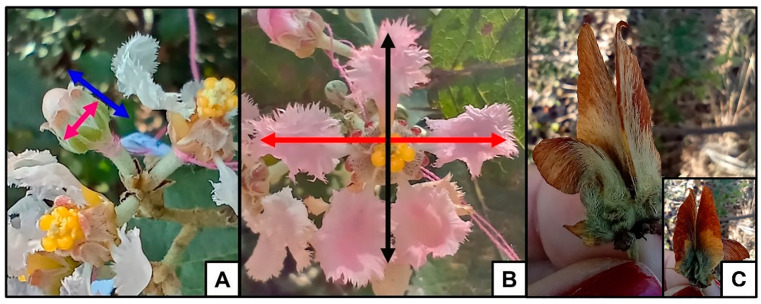
(**A**) Bud in pre-anthesis, the arrows indicate the directions that the height (blue arrow) and diameter (pink arrow) of the buds were measured. (**B**) Open flower, the arrows indicate how the width (red arrow) and length (black arrow) of the flowers were measured after anthesis. (**C**) Fruit (samaras). Photos by the author.

**Table 1 plants-14-00837-t001:** Ant species found on *Banisteriopsis malifolia* (Malpighiaceae) plants, without (Control) and with (T15, T50) simulated herbivory.

Species of Ants	Control		T15		T50	
	Aab ^1^*	Rab ^2^	Aab	Rab	Aab	Rab
*Azteca* sp.	0	-	2	14%	0	-
*Brachymyrmex* sp.	0	-	1	7%	0	-
*Camponotus* sp.	2	7%	2	14%	5	35%
*Dolichoderus* sp.	4	14%	1	7%	0	-
*Ectatomma tuberculatum* (Olivier, 1792)	16	59%	0	-	9	65%
*Ectatomma* sp.	2	7%	0	-	0	-
*Pseudomyrmex* sp.	3	10%	0	-	0	-
*Tapinoma* sp.	1	3%	2	14%	0	-
*Wasmannia* sp.	0	-	6	44%	0	-
**Total**	28		14		14	

^1^* absolute abundance, ^2^ relative abundance of ants.

**Table 2 plants-14-00837-t002:** Values of χ^2^, *p*, marginal R, and conditional R of each floral trait, and mean and confidence interval (CI) (95%) of floral traits and the variables between treatments in *Banisteriopsis malifolia* (Malpighiaceae) plants without (Control) and with simulated herbivory (T15, T50).

Floral Traits	χ^2^	*p*	R^2^m ^1^	R^2^c ^2^	Treatments	Mean	CI (95%)
							LL, UL ^3^
Flower bud heights	9.3719	0.009	0.13	0.58	Control	6.76	6.49, 7.03
					T15	6.50	6.24, 6.76
					T50	6.15	5.87, 6.43
Flower bud diameters	5.9038	0.05	0.09	0.61	Control	5.88	5.63, 6.14
					T15	5.46	5.22, 5.70
					T50	5.59	5.32, 5.85
Flower width	7.5617	0.02	0.13	0.48	Control	27.8	26.0, 29.5
					T15	24.6	22.9, 26.2
					T50	25.2	23.3, 27.0
Flower length	1.8328	0.399	0.03	0.54	Control	25.2	23.0, 27.4
					T15	23.2	21.2, 25.3
					T50	23.8	21.5, 26.1
Inflorescence size	13.232	0.001	0.09	0.15	Control	12.67	9.95, 15.38
					T15	14.98	11.77,18.18
					T50	6.44	2.91, 9.98
Flowers per inflorescence	9.333	0.009	0.14	0.86	Control	15.62	11.84, 20.6
					T15	11.19	8.32, 15.0
					T50	7.99	5.69, 11.2
Flowers by inflorescence size	5.988	0.05	0.05	0.25	Control	1.46	1.125, 1.80
					T15	1.16	0.774, 1.54
					T50	1.87	1.437, 2.30

^1^ Marginal R, ^2^ Conditional R, ^3^ LL (Lower limit), UL (Upper limit).

## Data Availability

The original data presented in the study are openly available in Figshare at https://doi.org/10.6084/m9.figshare.28243361.

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
