# Peer review of "Impacts of Leaf Damage Intensity on Ant–Plant Protection Mutualism and Plant Fitness"

_plants, 2025, doi:10.3390/plants14060837_

Round 1
Reviewer 1 Report
Comments and Suggestions for Authors
This paper is well-written and represents work that contributes to the understanding of plant-herbivore interactions. There are some instances where the text could be revised to improve clarity and where typos should be corrected. However, the majority of my comments relate to the interpretation of the results, especially in regards to treatment influence on ant abundance. This can be easily revised in the discussion without losing the relevance of this research. See the attached document for specific comments.

Author Response
Reviewer: #1
Dear Reviewer,
We would like to express our sincere gratitude for your valuable suggestions and for the relevant citation you shared. Your contributions have been essential in improving the quality and clarity of our manuscript.
We have carefully incorporated all your suggestions into the text, ensuring that the points raised were properly addressed. Below is a summary of the modifications made based on your recommendations:
- Citation added (L. 40).
- The order of chemical, physical, biotic, or phenological was adjusted as requested (L. 44).
- 'For instance' was changed to 'Relatedly' (L. 48).
- 'Combined' was changed to 'in combination' (L. 55).
- 'On' was corrected to 'one' (L. 58).
- The sentence "We conducted a field experiment using different levels of artificial damage to plants to assess how this plant responds EFN production, ant foraging patterns, and plant reproductive success, but the magnitude of these effects is directly related to the level of damage." was revised to "We conducted a field experiment using different levels of artificial damage to plants to assess how this plant responds EFN quality, ant foraging patterns, and plant reproductive success." (L. 77-79).
- 'And' was added before the third prediction (L. 86).
- The phrase "which increased in proportion to leaf damage" was removed (L. 98).
- The total number of ants found in each treatment and in the control was added to Table 1.
- The phrase "Since the treatments influence the sugar concentration of the nectar" was modified to "Since treatments T15 influenced the sugar concentration of the nectar" (L. 120).
- The phrase "this suggests that ant foraging is influenced by the herbivory presented in the treatments" was removed as requested for being redundant (L. 126).
- The word "controle" in Figure 10 was corrected.
- 'Production' was changed to 'quality' (L. 241).
- The graphical abstract (previously Figure 3) was modified.
- The passage "The increase in 15% of leaf herbivory induced different responses in the plants, with a greater investment in increasing the sugar concentration of the EFNs, consequently attracting more ants (prediction ii) [36]. Although there was no statistical difference between the sugar concentration between Control and T50 plants, T50 plants produced 1.37 times more sugar than Control, which might explain the increase in the number of ants attracted in this treatment under herbivory." was revised to "The increase in 15% of leaf herbivory induced different responses in the plants, with a greater investment in increasing the sugar concentration of the EFNs [35]. Although there was no statistical difference in sugar concentration between the Control and T50 plants, T50 plants produced 1.37 times more sugar than the Control plants, which may explain why only more aggressive ants appeared in T50 compared to the other treatments (Table 1) [37]. Ant protection can reduce herbivory, allowing the plant to allocate more resources to its growth and reproduction, resulting in increased production of leaves, flowers, fruits, and seeds." (L. 250-257).
- "And attract more protective ants" was removed (L. 302).
- The description of C in Figure 9 (previously Figure 2) was added.
- 'Samarids' was corrected to 'samaras' (L. 400).
- "Since the volume of nectar did not differ between treatments" was removed (L. 421).
- 'Secretion' was changed to 'sugar concentration' (L. 461).
- The sentence "In contrast, severe damage (T50) leads to the prioritization of reproduction over defense, negatively impacting reproductive performance." was modified to "While severe damage (T50) negatively impacts reproductive performance." (L. 464).
Response #5: Regarding the comment, "Since the detection ability was capped at 50, it's possible there could be a bimodal distribution here, which would represent interesting physiological relevance and implications," in Figure 1 (previously Figure 4), the very high values could be due to nectar water evaporation, leading to higher concentrations. This may occur despite our rigorous adherence to the methodology, potentially as a result of low nectar production rather than necessarily indicating a bimodal distribution. However, we agree that this is a great point and should be more investigated.
Reviewer 2 Report
Comments and Suggestions for Authors
This interesting paper studies the relationship between plant physiology and herbivory attack. Although the research follows a logical flow of results, some points need further clarification to make the paper more accessible for non-specialist readers.
a) The experimental design uses wounding or mechanical stress (cutting with scissors) to simulate an herbivore attack. The point is that it is known that plants can detect chemical signals from herbivore insects and elicit different responses. With this approach, these signals are not present. Is this design ubiquitous in insect-plant studies? Can the authors cite studies with similar designs?
b) Related to the previous one. The study was not conducted in a greenhouse or a plant growth chamber under controlled conditions but in a natural environment. Are there any insects present? How could the authors control the presence of insects that could alter the results of the study?
Author Response
Reviewer: #2
- a) The experimental design uses wounding or mechanical stress (cutting with scissors) to simulate an herbivore attack. The point is that it is known that plants can detect chemical signals from herbivore insects and elicit different responses. With this approach, these signals are not present. Is this design ubiquitous in insect-plant studies? Can the authors cite studies with similar designs?
Response # 6: You are correct, this study does not show the chemical signals. However, the simulation of mechanical damage is still widely used, as it allows for investigating plant responses to physical damage in a controlled manner. Our lab has conducted studies with this approach (Calixto et al. 2021, 2023), which has also been used for other researchers (Gray et al. 2024 (doi:10.1093/aob/mcae118), Mao et al. 2022 (doi: 10.3389/fpls.2022.902342), Pulice and Packer 2008, Raupp et al. 2020). We added more citations in the manuscript to support our methods (L. 38,263)
- b) Related to the previous one. The study was not conducted in a greenhouse or a plant growth chamber under controlled conditions but in a natural environment. Are there any insects present? How could the authors control the presence of insects that could alter the results of the study?
Response #7: Yes, there were insects present in the natural environment of the study. However, we indicated in the article that herbivory in the control was below 5% (abstract: L. 17 and L. 333), suggesting a minimal impact of insects on the results. In addition, we used many plants per treatment to reduce the bias of any plant getting attacked and altering the results.
Reviewer 3 Report
Comments and Suggestions for Authors
In this MS, a series of experiments were carried out to study the impacts of leaf damage intensity on ant-plant protection mutualism and plant fitness. The topic is very interesting and hot issues about ant-plant mutualism. While there are many shortcomings which were following as:
Q1: In the title of this MS, the topic of ant-plant protection mutualism was shown, while how about the protection or protection mutualism were not studied, just the abundance of ants were measured under the simulated damage (T15 and T50 vs Control).
Q2: Many subtitles of this MS were not suitable not to indicate the measured indexes or study objectives, e.g., 2.1. Extrafloral Nectaries and Ants: Not clear subtitle to indicate the measured indexes of plants and ants under different damage treatments; 2.2. Floral traits: Not clear subtitle to indicate the measured indexes; 4.1. Study area and species evaluated: Species evaluated? How to evaluate species of plants? Not suitable subtitle! 4.6. Statistical analysis: Statistical Analysis should be at the end of the M&M section. Moreover, some repeated subtitles were given, e.g., the subtitles of 4.3 and 4.7, those of 4.4 and 4.8, those of 4.5 and 4.9.
Q3: Abstract: No data results were given about ant foraging behavior in the Abstract. What about the effects of ant protection on plants against insect herbivores' damage? Not given!
Q4: 2.1: What about that R²m and R²c? Not introduced clear!
Q5: Figure 4 not Figure 1 was firstly numbered, while Table 1 not Table 2 was firstly numbered in this MS.
Q6: Figure 5: What about the three types of color data points for Control, T15 and T50?

Author Response
Reviewer: #3
In this MS, a series of experiments were carried out to study the impacts of leaf damage intensity on ant-plant protection mutualism and plant fitness. The topic is very interesting and hot issues about ant-plant mutualism.
Response #8: Thank you for your time and suggestions, which we have addressed in this new version of the manuscript.
Q1: In the title of this MS, the topic of ant-plant protection mutualism was shown, while how about the protection or protection mutualism were not studied, just the abundance of ants were measured under the simulated damage (T15 and T50 vs Control).
Response #9: Our study evaluates the impact of a protection mutualism between ants and plants but does not directly measure the protective effect itself. Instead, we assume the interaction is already classified as a protection mutualism, which is supported by many studies with this plant, (Calixto et al. 2021 (doi:10.1111/1365-2745.13457); Alves-Silva 2011 Sociobiology; Assunção, Torezan-Silingardi and Del-Claro 2014 (doi:10.1016/j.flora.2014.03.003) and Del-Claro et al. 2016 (doi: 10.1007/s00040-016-0466-2)). Therefore, the term in the title refers to the general nature of the interaction rather than a direct test of the protection provided by the ants.
Q2: Many subtitles of this MS were not suitable not to indicate the measured indexes or study objectives, e.g., 2.1. Extrafloral Nectaries and Ants: Not clear subtitle to indicate the measured indexes of plants and ants under different damage treatments; 2.2. Floral traits: Not clear subtitle to indicate the measured indexes; 4.1. Study area and species evaluated: Species evaluated? How to evaluate species of plants? Not suitable subtitle! 4.6. Statistical analysis: Statistical Analysis should be at the end of the M&M section. Moreover, some repeated subtitles were given, e.g., the subtitles of 4.3 and 4.7, those of 4.4 and 4.8, those of 4.5 and 4.9.
Response #10: All subtitles have been modified as requested. The statistical analysis is included at the end of the M&M section. "Study area and species evaluated" has been changed to "Study area and species studied."
Q3: Abstract: No data results were given about ant foraging behavior in the Abstract. What about the effects of ant protection on plants against insect herbivores' damage? Not given!
Response #11: The results on ant behavior have been included in the abstract (L. 22-23) and addressed in the discussion (L. 250-252).
Q4: 2.1: What about that R²m and R²c? Not introduced clear!
Response #12: Thank you for pointing this out. R² marginal" refers to the proportion of variance explained by the fixed effects alone, while "R² conditional" represents the proportion of variance explained by both the fixed and random effects in the model; essentially, the marginal R² only considers the fixed effects, while the conditional R² takes into account both fixed and random effects. We have now added this in the text (L. 418-421)
Q5: Figure 4 not Figure 1 was firstly numbered, while Table 1 not Table 2 was firstly numbered in this MS.
Response #13: Thank you for your observation. We have revised the manuscript and adjusted the numbering order so that figures and tables are presented sequentially.
Q6: Figure 5: What about the three types of color data points for Control, T15 and T50?
Response #14: The caption has been added to the figure for clarification.